# Understanding Generalization in Transformers: Error Bounds and Training Dynamics Under Benign and Harmful Overfitting

## Abstract

Transformers serve as the foundational architecture for many successful large-scale models, demonstrating the ability to overfit the training data while maintaining strong generalization on unseen data, a phenomenon known as benign overfitting. However, existing research has not sufficiently explored generalization and training dynamics of transformers under benign overfitting. This paper addresses this gap by analyzing a two-layer transformer's training dynamics, convergence, and generalization under labeled noise. Specifically, we present generalization error bounds for benign and harmful overfitting under varying signal-to-noise ratios (SNR), where the training dynamics are categorized into three distinct stages, each with its corresponding error bounds. Additionally, we conduct extensive experiments to identify key factors in transformers that influence test losses. Our experimental results align closely with the theoretical predictions, validating our findings.

## 1 Introduction

In recent years, benign overfitting has reshaped our understanding of overparameterization in deep neural networks. Traditional viewpoints hold that models with more parameters than training samples tend to overfit, resulting in poor generalization performance on new data. However, modern deep neural networks challenge this viewpoint by demonstrating remarkable generalization capabilities. Despite having sufficient parameters to perfectly fit training data, they still maintain low test loss Zhang et al. (2017); Neyshabur et al. (2018). This phenomenon, known as benign overfitting, has attracted significant attention across both statistical and machine learning communities Belkin et al. (2018; 2019; 2020); Neyshabur et al. (2018); Hastie et al. (2022).

Researchers have investigated benign overfitting from conventional perspective, while these works are related to linear models Chatterjee & Long (2022); Zou et al. (2021), kernel methods or random feature models Montanari & Zhong (2022); Adlam & Pennington (2020); Zhu Li (2021). Researchers have expanded these theoretical analyses to study benign overfitting in neural networks Adlam & Pennington (2020); Zhu Li (2021). They are still limited to the neural tagent kernel regime (NTK) Jacot et al. (2018) because the neural network learning problem is equivalent to kernel regression. Several works further study benign overfitting and generalization in transformers. These analyses typically focus on simplified settings, such as linear transformers Frei & Vardi (2024). Yet, due to the self-attention mechanism and softmax activation function, the transformer exhibits nonlinear learning in the real world, rendering the above simplifying assumption unreasonable.

Recent theoretical works have studied the benign overfitting and generalization of transformers with nonlinear self-attention Jiang et al. (2024); Magen et al. (2024), and some even have extended to context learning tasks Li et al. (2024b). Our analysis of benign overfitting and generalization in transformers is compared to existing research, as summarized in Table 1. However, several studies Frei & Vardi (2024); Li et al. (2024a) only considered generalization in a single type of overfitting (either benign or harmful). Others Jiang et al. (2024); Li et al. (2024a) analyzed the generalization of transformers under the assumption of clean data labels, which is unreasonable in real world. Therefore, an important open question remains:

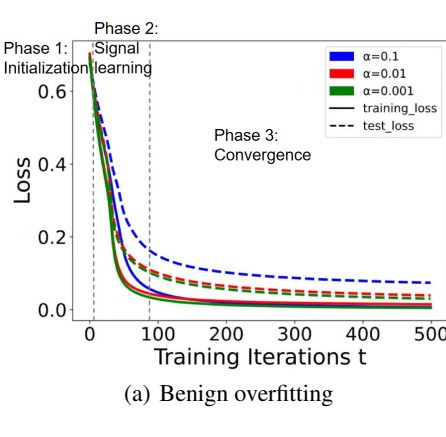 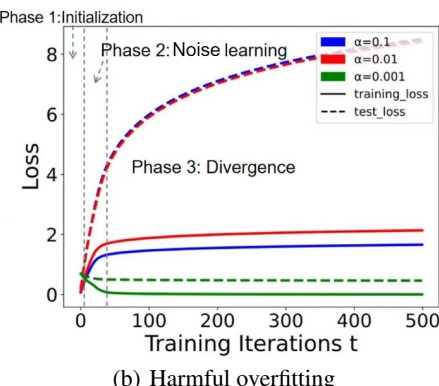

(a) Benign overfitting        (b) Harmful overfitting

Figure 1: Test losses of benign overfitting and harmful overfitting under label noise (parameterized by $\alpha$).

*How do transformers generalize under labeled noise while considering both benign overfitting and harmful overfitting?*

Our work aims to settle down the above question through feature learning framework by analyzing a two-layer transformer's training dynamics, convergence, and generalization under labeled noise. Specifically, we consider two tokens including signal and noise, and a two-layer nonlinear transformer with softmax activation function. We explore the training dynamics of transformers in both benign overfitting and harmful overfitting, and provide corresponding error bounds. The theoretical bounds illutrate three distinct stages for benign overfitting and harmful overfitting, respectively. We then conduct extensive experiments to validate our theoretical finding. As shown in Figure 1, the test losses for benign overfitting and harmful overfitting divide into three distinct stages and the empirical loss is upper bounded by the theoretical bound (in Figure 2).

| Theoretical Works | Nonlinear | Labeled Noise | Benign Overfitting | Harmful Overfitting | Stage-wise Error Bounds |
|---|---|---|---|---|---|
| Li et al. (2024a) | ✓ | ✗ | ✗ | ✓ | ✗ |
| Sakamoto & Sato (2024) | ✓ | ✓ | ✓ | ✓ | ✗ |
| Jiang et al. (2024) | ✓ | ✗ | ✓ | ✓ | ✗ |
| Frei & Vardi (2024) | ✗ | ✓ | ✓ | ✗ | ✗ |
| Magen et al. (2024) | ✓ | ✓ | ✓ | ✓ | ✗ |
| This work | ✓ | ✓ | ✓ | ✓ | ✓ |

Table 1: Theoretical Comparison with existing works on benign overfitting and generalization.

Our contributions are summarized as follows:

- **Theoretical Contribution I :** We consider a nonlinear transformer with softmax activation function. Additionally, we relax the assumption of clean data labels and incorporate labeled noise to more accurately reflect real-world conditions.

- **Theoretical Contribution II :** We examine the training dynamics of transformers under labeled noise in both benign overfitting and harmful overfitting. The training dynamics associated with benign overfitting can be characterized by three distinct phases: **initialization**, **signal learning**, and **convergence**. In contrast, harmful overfitting is characterized by **initialization**, **noise learning**, and **divergence**. In Theorem 1 and Theorem 2, we provide specific stage-wise error bounds for each phase.

- **Experimental Contribution :** We investigate the transition between benign overfitting and harmful overfitting. Additionally, to further enhance the model's generalization perfor-

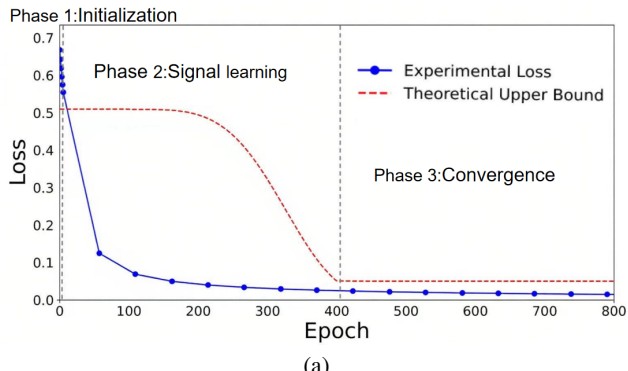

Figure 2: Numerical comparison between the theoretical upper bound and the experimental loss for benign overfitting.

mance, we analyze several key factors relevant to generalization during benign overfitting. These experimental results validate our theoretical analysis.

## 2 RELATED WORK

### 2.1 BENIGN OVERFITTING IN TRADITIONAL MODELS.

Several works explored benign overfitting in traditional models, including linear models Bartlett et al. (2020); Zou et al. (2021); Cao et al. (2021); Mo Zhou (2023), kernel methods, and random feature architectures. Zou et al. (2021) derived excess risk bounds for stochastic gradient descent with constant step sizes. Liao et al. (2021) expanded the analysis to random Fourier feature regression, focusing on fixed asymptotic ratios of sample size, data dimensionality, and feature count. As shown in Liang & Rakhlin (2020); Adlam & Pennington (2020); Zhu Li (2021); Montanari & Zhong (2022); Chatterjee & Long (2022); Spencer Frei (2022), several studies have broadened conventional perspectives to investigate benign overfitting in neural networks based on traditional models. The authors in Adlam et al. (2021) explored a precise analysis of generalization under nuclear regression, while Tsigler & Bartlett (2022) demonstrated that overparameterized ridge regression models can achieve benign overfitting even when fitting noisy data, and extended this to ridge regression conditions. Mallinar et al. (2024) discovered that interrupting training prematurely in neural networks leads to benign overfitting, while deep neural networks trained to full interpolation do not exhibit this phenomenon. Unlike these research, our work focuses on benign overfitting in transformers, which is more challenging than neural networks.

### 2.2 BENIGN OVERFITTING IN TRANSFORMER.

Towards understanding the benign overfitting and generalization in transformers, Frei & Vardi (2024) investigated the behavior of linear transformers trained on random linear classification tasks and quantifies how many examples transformers need in context learning to generalize well. Building on this, Magen et al. (2024) investigated benign overfitting in single-head attention models, revealing that this phenomenon only occurs when the signal-to-noise ratio reaches a sufficiently high level, and Sakamoto & Sato (2024) further explored benign overfitting in the token selection mechanism of the attention. The work in Li et al. (2024a) investigated the training dynamics of harmful overfitting when optimizing two-layer transformers using symbolic gradient descent. Most relevant to our work is Jiang et al. (2024), as they also study the benign overfitting and generalization of transformer with a similar data model. However, they do not take into account the effect of labeled noise, which is more common and realistic in real-world. In this paper, we bridge this gap by analyzing the generalization of transformers in benign overfitting and harmful overfitting under labeled noise condition.

## 3 PROBLEM SETUP

In this section, we denote the data generation model, two-layer transformer model, and the gradient descent training algorithm.

**Notions.** We define two sequences $\{a_n\}$ and $\{b_n\}$, which have the following relationship. We define $a_n = O(b_n)$ and $b_n = \Omega(a_n)$ if there exist $|a_n| \leq c_1 |b_n|$ for some positive constant $c_1$. At the same time, we define $a_n = \Theta(b_n)$ if $a_n = O(b_n)$ and $a_n = \Omega(b_n)$ hold.

**Definition 1.** Let $\boldsymbol{\mu}_+, \boldsymbol{\mu}_- \in \mathbb{R}^d$ be fixed vectors which represent the signals contained in each data point $(\mathbf{X}, y)$, where $\|\boldsymbol{\mu}_+\|_2 = \|\boldsymbol{\mu}_-\|_2 = \|\boldsymbol{\mu}\|_2$ and $\langle \boldsymbol{\mu}_+, \boldsymbol{\mu}_- \rangle = 0$. Then we define each data point $(\mathbf{X}, y)$ with the input features $\mathbf{X} = (\boldsymbol{x}_1, \boldsymbol{x}_2) \in \mathbb{R}^{d \times 2}$, and $y \in \{\pm 1\}$ is generated from the model:

- True labels $\hat{y} \in \{\pm 1\}$ are Rademacher random variables with $\mathbb{P}[\hat{y} = 1] = \mathbb{P}[\hat{y} = -1] = 1/2$. Observed labels $y$ are generated by flipping $\hat{y}$ with probability $\alpha$, i.e., $\mathbb{P}[y = \hat{y}] = 1 - \alpha$ and $\mathbb{P}[y = -\hat{y}] = \alpha$.

- The signal vector $\boldsymbol{x}_1$ is denoted $\boldsymbol{\mu}_+$ if $\hat{y} = 1$, and $\boldsymbol{\mu}_-$ if $\hat{y} = -1$.

- The noise vector $\boldsymbol{x}_2 = \boldsymbol{\xi}$ is sampled from $\boldsymbol{\xi} \sim \mathcal{N}(0, \sigma_p^2 \boldsymbol{I}_d)$.

We consider each data point as a vector of two tokens, $\mathbf{X} = (\boldsymbol{x}_1, \boldsymbol{x}_2)^T \in \mathbb{R}^{2 \times d}$. The token $\boldsymbol{x}_1$, represents the signal that is inherently linked to the data's true class label, such as $\boldsymbol{\mu}_+$ and $\boldsymbol{\mu}_-$, while $\boldsymbol{x}_2$, serves as noise and is irrelevant to the label. Building on Definition 3.1 from Jiang et al. (2024), we further refine the data distribution to enhance its practical applicability. Specifically, we introduce label-flipping noise to the true label $\hat{y}$.

**Signal-to-Noise Ratio (SNR).** From Cao et al. (2022), when the dimension $d$ is large, the norm of the noise vector satisfies $\|\boldsymbol{\xi}\|_2 \approx \sigma_p \sqrt{d}$ based on standard concentration bounds. Therefore, the signal-to-noise ratio (SNR) can be expressed as $\mathrm{SNR} \approx \|\boldsymbol{\mu}\|_2 / \sigma_\mathrm{p} \sqrt{\mathrm{d}}$, which is approximately equal to $\|\boldsymbol{\mu}\|_2 / \|\boldsymbol{\xi}\|_2$. Hence, we use the expression $\mathrm{SNR} \approx \|\boldsymbol{\mu}\|_2 / \sigma_\mathrm{p} \sqrt{\mathrm{d}}$ to represent the signal-to-noise ratio.

**Two-layer Transformer.** We define the model as a two-layer transformer, consisting of an attention layer with softmax activation function and a fixed linear layer. Let $\mathbf{S}$ represent the softmax function. we categorize the output of the softmax function into four types of vectors, corresponding to the softmax outputs of the pairwise inner products involving the query signal, query noise, key signal, and key noise.

Specifically, the signal-to-signal output $S_{11}$, signal-to-noise output $S_{12}$, noise-to-signal output $S_{21}$, and noise-to-noise output $S_{22}$ have been defined in the supplementary material. For example, the signal-to-signal output $S_{11}$ can be written as:

$$\mathbf{S}_{11} = \mathrm{Softmax}(\langle q_\pm^{(t)}, k_\pm^{(t)} \rangle) = \begin{cases} \frac{\exp(\langle q_+, k_+ \rangle)}{\exp(\langle q_+, k_+ \rangle) + \exp(\langle q_+, k_{\xi,i} \rangle)} & \text{for } i \in [\mathbf{S}_+], \\ \frac{\exp(\langle q_-, k_- \rangle)}{\exp(\langle q_-, k_- \rangle) + \exp(\langle q_-, k_{\xi,i} \rangle)} & \text{for } i \in [\mathbf{S}_-]. \end{cases}$$

Let $\mathbf{S}_+$ be the set of indices $i$ in $[N]$ where $y_i = 1$, and let $\mathbf{S}_-$ be the set of indices $i$ in $[N]$ where $y_i = -1$. Note that $q_+$, $k_+$, $q_-$, $k_-$, and $k_{\xi,i}$ are related to the query with $+1$ label, the key with $+1$ label, the query with $-1$ label, the key with $-1$ label, and the key with noise, respectively. The output result can be given as: $f(\mathbf{X}, \upsilon) = f_{+1}(\mathbf{X}, \upsilon) - f_{-1}(\mathbf{X}, \upsilon)$, where $f_j(\mathbf{X}, \upsilon)$ for $j \in \{\pm 1\}$ is defined as:

$$f_j(\mathbf{X}, \upsilon) = \sum_{l=1}^{2} \upsilon^\top \mathbf{W}_{V,j}^\top \mathbf{X} \mathbf{S} \left( \mathbf{X}^\top \mathbf{W}_Q \mathbf{W}_K^\top \boldsymbol{x}_l \right) = \sum_{l=1}^{2} \upsilon^\top \left( \sum_{r=1}^{d_V} \mathbf{W}_{Vj,r}^\top \mathbf{X} \right) \mathbf{S} \left( \mathbf{X}^\top \mathbf{W}_Q \mathbf{W}_K^\top \boldsymbol{x}_l \right).$$

The parameter of the linear layer is denoted as $\upsilon \in \mathbb{R}^{d_V}$. The parameters of the attention layer are defined as $\mathbf{W}_Q, \mathbf{W}_K, \mathbf{W}_{V,j}$, where $\mathbf{W}_Q, \mathbf{W}_K \in \mathbb{R}^{d \times d_K}$ and $\mathbf{W}_{V,j} \in \mathbb{R}^{d \times d_V}$, representing the query matrix, the key matrix, and the value matrix respectively. We use $\theta$ to represent all the parameters of the attention model, which is defined as $\theta = (\mathbf{W}_Q, \mathbf{W}_K, \mathbf{W}_{V,j})$. We rewrite the model in a specific form for $j \in \{\pm 1\}$:

$$f_j(\theta, \mathbf{X}, \upsilon) = \sum_{r \in [d_V]} \left( \upsilon^\top \langle \mathbf{W}_{Vj,r}, \boldsymbol{x}_1 \rangle (\mathbf{S}_{11} + \mathbf{S}_{21}) + \upsilon^\top \langle \mathbf{W}_{Vj,r}, \boldsymbol{x}_2 \rangle (\mathbf{S}_{12} + \mathbf{S}_{22}) \right).$$

**Training Algorithm.** We use a training dataset $S = \{(\mathbf{X}_i, y_i)\}_{i=1}^N$ generated from the distribution $D$ defined in Definition 1. Our transformer model is trained by minimizing the logistic loss function: $L_S(\theta) = \frac{1}{N} \sum_{i=1}^N \ell(y_i f(\theta, \mathbf{X}, v))$, where $\ell(z) = \log(1 + \exp(-z))$. We employ gradient descent to minimize the training loss $L_S(\theta)$, and focus on characterizing the test error(i.e., true error), defined by: $L_{\mathcal{D}}^{0-1}(\theta) = \mathbb{P}_{(\mathbf{x},y) \sim \mathcal{D}}[y \neq \text{sign}(f(\theta, \mathbf{X}, v))]$. For the sake of simplification, we consider gradient descent optimization, and we have $\mathbf{W}_V^{(t+1)} = \mathbf{W}_V^{(t)} - \eta \left(\nabla_{\mathbf{W}_V} L_S(\mathbf{W}^{(t)})\right)$, $\mathbf{W}_Q^{(t+1)} = \mathbf{W}_Q^{(t)} - \eta \left(\nabla_{\mathbf{W}_Q} L_S(\mathbf{W}^{(t)})\right)$, and $\mathbf{W}_K^{(t+1)} = \mathbf{W}_K^{(t)} - \eta \left(\nabla_{\mathbf{W}_K} L_S(\mathbf{W}^{(t)})\right)$.

# 4 MAIN RESULTS

In this section, we present our main theoretical findings. These findings are based on several key conditions as follows:

**Assumptions .** Given a sufficiently small failure probability $\delta > 0$, a large constant $c_1$, and a target training loss $\epsilon > 0$, suppose the following conditions hold:

(1) The dimension $d_K$ satisfies: $d_K \geq \begin{cases} \text{SNR}^4 N^4 \epsilon^{-4}, & \text{if } \|\mu\| \geq \sigma_p \sqrt{d}, \\ \text{SNR}^{-4} N^4 \epsilon^{-4}, & \text{if } \|\mu\| < \sigma_p \sqrt{d}. \end{cases}$

(2) The dimension $d$ satisfies: $d \geq \text{poly}(d_K)$.

(3) The training sample size $N$ satisfies: $N \geq c_1 \cdot \text{polylog}(d)$.

(4) The label-flipping probability $\alpha$ satisfies: $\alpha \in [0, 1/2)$.

(5) The linear layer weight satisfies: $\|v\|_2 = \Theta(1)$.

(6) The learning rate $\eta$ satisfies: $\eta \leq O\left(\min\left\{\sigma_p^2 d, \|\boldsymbol{\mu}\|_2^2\right\} N^2 \epsilon^{-2}\right)$.

(7) The parameters $\mathbf{W}_Q$ and $\mathbf{W}_K$ are initialized from a Gaussian distributions $\mathcal{N}(0, \sigma_K^2)$ and the variance satisfies: $\sigma_K^2 \leq O\left(\max\left\{(\sigma_p^2 d)^{-1}, \|\boldsymbol{\mu}\|_2^{-2}\right\} N^{-1} \epsilon \log \frac{24N^2}{\delta}\right)^{-3/2}$, while $\mathbf{W}_V$ is initialized from $\mathcal{N}(0, \sigma_V^2)$ where

$$\sigma_V \leq \begin{cases} O\left(\frac{\sqrt{\epsilon}}{\sqrt{dN}\|v\|\sigma_p}\right), & \text{if } \|\mu\| \geq \sigma_p \sqrt{d}, \\ O\left(\frac{\sqrt{\epsilon}}{\sqrt{N}\|v\|\|\mu\|}\right), & \text{if } \|\mu\| < \sigma_p \sqrt{d}. \end{cases}$$

Assumptions (1)–(3) ensure that the transformer operates in an over-parameterized setting. Similar assumptions have been made in neural networks Cao et al. (2022); Kou et al. (2023). Noise in training data is common in real-world environments. To address this gap, we relax the assumption of clean data labels and incorporate labeled noise $\alpha$ to more accurately reflect real-world conditions. As a result, the generalization error bound derived under this assumption is more meaningful in practice. Assumption (4) ensures that we do not incorporate excessive noise, which could significantly impair the transformer's generalization. This assumption is frequently used in theoretical analyses Kou et al. (2023); Frei & Vardi (2024); Sakamoto & Sato (2024). Assumption (5) is realistic in practice, as it controls the range of weights through appropriate training strategies. Assumptions (6)–(7) ensure that gradient descent can effectively minimize the training loss. Similar assumptions have been widely used in feature learning theories Cao et al. (2022); Jiang et al. (2024).

**Theorem 1 (Benign overfitting in transformers).** *When $N \cdot SNR^2 + h(\alpha) = \Omega(1)$, where $h(\alpha)$ is a function related to $\alpha$, for any $\epsilon > 0$, under the assumptions above, with probability at least $1 - \delta$:*

- *(Phase 1: Initialization) There exists $T_1 = O\left(\frac{1}{\eta d_K^{\frac{1}{4}}\|\boldsymbol{\mu}\|_2^2\|v\|_2^2}\right)$, and for $t \in (0, T_1]$, the test loss is upper bounded by:*

$$L_{\mathcal{D}}^{0-1}(\theta(t)) \leq \frac{1}{2} + \alpha + \mathcal{O}(1).$$

- **(*Phase 2: Signal learning*)** *There exists* $T_2 = \Theta\left(\frac{1}{\eta\|\mu\|_2^2\|v\|_2^2}\right)$, *for* $t \in (T_1, T_2]$, *the test loss is upper bounded by:*

$$L_{\mathcal{D}}^{0-1}(\theta(t)) \lesssim \alpha + \exp\left(-\eta^4\|\mu\|_2^8(t-T_1)^4 SNR^2\right).$$

- **(*Phase 3: Convergence*)** *There exists* $t > T_2$ *such that:*
  - *The training loss converges to* $\epsilon$: $L_S(\theta(t)) \leq \epsilon$.
  - *The test loss is upper bounded by:*

$$L_{\mathcal{D}}^{0-1}(\theta(t)) \lesssim \alpha + \exp\left(-\frac{\eta^4(t-T_2)^4\|\mu\|_2^6 \cdot SNR^2}{\sigma_V^2}\right).$$

Theorem 1 illustrates the generalization behavior of transformers under benign overfitting when $N \cdot \mathrm{SNR}^2 + h(\alpha) = \Omega(1)$. Under this condition, the error bounds of transformers can be divided into three distinct phases:

- **Initialization phase**: Initially, the transformer parameters have not been adequately trained, leading to a test loss that remains at a significant constant level of $\Omega(1)$. This phase is primarily influenced by $\alpha$ and the random initialization parameters ($\sigma_V$ and $\sigma_K^2$).
- **Signal learning phase**: During this phase, the model focuses more on learning the signals rather than the noises, which results in an increase in test loss. The test loss is governed by an upper bound that is directly proportional to time $t$, the labeled noise $\alpha$, the learning rate $\eta$, the signal strength $\|\mu\|$, and the square of the signal-to-noise ratio $\mathrm{SNR}^2$.
- **Convergence phase**: When $t > T_2$, the training loss converges to a low level $\epsilon$. In this phase, the upper bound of the test loss is influenced by several key factors, including time $t$, the labeled noise $\alpha$, the learning rate $\eta$, the signal strength $\|\mu\|$, and $\mathrm{SNR}^2$. Notably, it is inversely proportional to the initialization variance $\sigma_V^2$. By carefully tuning these factors under benign overfitting condition, we can achieve a lower test loss, which is the primary objective of our work in this paper.

**Theorem 2** (**Harmful overfitting in transformers**). *When* $N^{-1} \cdot SNR^{-2} + h(\alpha) = \Omega(1)$, *where* $h(\alpha)$ *is a function related to* $\alpha$, *for any* $\epsilon > 0$, *under the assumptions, with probability at least* $1-\delta$:

- **(*Phase 1: Initialization*)** *There exists* $T_1 = O\left(\frac{N}{\eta d_K^{\frac{1}{4}}\|\mu\|_2^2\|v\|_2^2}\right)$, *for* $t \in (0, T_1]$, *such that the test loss is upper bounded by:* $L_{\mathcal{D}}^{0-1}((\theta(t)) \leq \frac{1}{2} + \alpha + O(1)$.

- **(*Phase 2: Noise learning*)** *There exists* $T_2 = \Theta\left(\frac{N}{\eta\sigma_p^2 d\|v\|_2^2 \log(24N^2/\delta)}\right)$. *For* $t \in (T_1, T_2]$, *the test loss is bounded by:*

$$L_{\mathcal{D}}^{0-1}(\theta(t)) \leq \frac{1}{2} + \alpha + O\left(\frac{1}{\|\mu\|_2^2\|v\|_2^2} + \frac{1}{\|\mu\|_2^4\|v\|_2^4}\right)$$

$$L_{\mathcal{D}}^{0-1}(\theta(t)) \geq \frac{1}{2} - O\left(\frac{1}{\|\mu\|_2^2\|v\|_2^4}\right).$$

- **(*Phase 3: Divergence*)** *There exists* $t > T_2$ *such that:*
  - *The training loss is higher than* $\epsilon$: $L_S((\theta(t)) \geq \epsilon$.
  - *The test loss is high:* $L_{\mathcal{D}}^{0-1}((\theta(t)) \geq \frac{1}{2}$.

Theorem 2 characterizes the generalization behavior of the transformer in harmful overfitting when $N^{-1} \cdot \mathrm{SNR}^{-2} + h(\alpha) = \Omega(1)$. The error bounds can be divided into three distinct phases:

- **Initialization phase**: Initially, the transformer parameters have not been sufficiently trained, resulting in the test loss remaining at a large constant value $\Omega(1)$. This is primarily influenced by $\alpha$ and random initialization ($\sigma_V$ and $\sigma_K^2$). This indicates that the model has not yet effectively learned the signals or the noises.

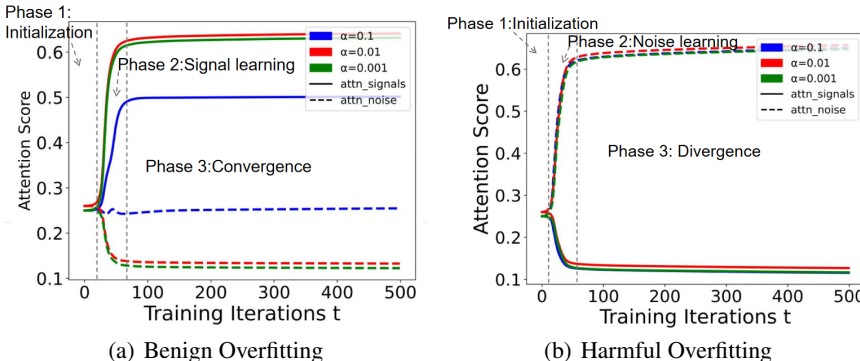

(a) Benign Overfitting

(b) Harmful Overfitting

Figure 3: Attention score analysis of benign overfitting and harmful overfitting under various labeled noise $\alpha \in \{0.2, 0.1, 0.01, 0.001\}$. We denote atten_signals as the strength of the signals learned by attention, while atten_noises represents the strength of the noise learned by attention.

- **Noise learning phase**: During this phase, the model increasingly focuses on the noises rather than the signals, leading to an increase in test loss. The test loss is upper bounded by a function directly related to the label-flipping noise $\alpha$, the signal strength $|\boldsymbol{\mu}|$, and the norm of the linear layer weight $\|v\|_2$. In contrast, the lower bound of the test loss is influenced solely by the signal strength $|\boldsymbol{\mu}|$ and the norm of the linear layer weight $\|v\|_2$.

- **Divergence phase**: When $t > T_2$, the model fully learns the noises. The test loss increases significantly and begins to diverge, ultimately exceeding $1/2$. This is higher than what would be expected from a random guess.

**Remark 1.** In summary, the model mainly learns the signals when benign overfitting occurs, resulting in lower loss values and better generalization. In contrast, when harmful overfitting occurs, the model mainly focuses on the noises, leading to poor generalization.

## 5 EXPERIMENTS

We present simulations using synthetic data to support our theoretical analysis. In this section, we demonstrate that the training dynamics can be clearly divided into three distinct phases based on varying $\alpha$ values across both overfitting scenarios. Furthermore, we confirm the existence of benign overfitting and investigate the conditions under which it occurs. Finally, we investigated methods to further enhance the model's generalization performance when benign overfitting occurs.

Synthetic data setting: We generate the training and test datasets according to Definition 1. Each data point consists of two components: signal and noise. The signal is composed of two orthogonal vectors, $\|\boldsymbol{\mu}\|_2 \cdot \mathbf{e}_1$ and $\|\boldsymbol{\mu}\|_2 \cdot \mathbf{e}_2$, which are generated with equal probability. $\mathbf{e}_1$ and $\mathbf{e}_2$ are defined as $[1, 0, \ldots, 0]^\top$ and $[0, 1, \ldots, 0]^\top$ respectively. The noise is sampled from a Gaussian distribution $\mathcal{N}(0, \sigma_p^2 \mathbf{I})$. In our experiments, the sample size $N$ is variable. Specifically, in the training dynamics and learning rate $\eta$ experiments, we vary $N$ from 2 to 20. In other experiments, we set $N$ to 100 to ensure the model learns the data sufficiently. Furthermore, to investigate the effect of signal-to-noise ratio (SNR) on benign overfitting, we adjust the signal strength $\mu$ from 1 to 100, while keeping the noise standard deviation $\sigma_p$ constant at 4. This allows us to explore how varying SNR impacts the test loss.

### 5.1 TRAINING DYNAMICS OF BENIGN OVERFITTING AND HARMFUL OVERFITTING

We primarily illustrate the training dynamics by examining the attention scores and the values of the $W_V$ matrix under various label-flipping noise conditions, encompassing both benign and harmful overfitting. Figure 3 (a) and (b) demonstrate that the training dynamics of attention can be characterized by three distinct phases. During the initialization phase, attention treats signals and noises equally, as it cannot distinguish between them. In the signal learning phase, attention increasingly focuses on the signals rather than the noises, and in the convergence phase, attention is entirely

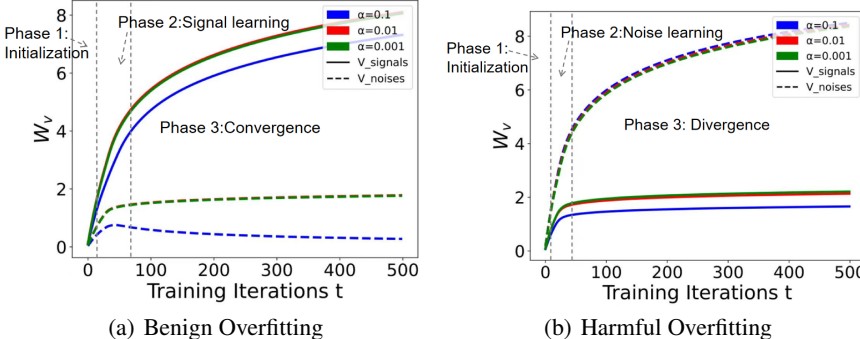

(a) Benign Overfitting            (b) Harmful Overfitting

Figure 4: Analysis of the $W_V$ matrix of benign overfitting and harmful overfitting under various labeled noise $\alpha \in \{0.2, 0.1, 0.01, 0.001\}$. We denote V_signals as the strength of the signals learned by the $\mathbf{W}_V$ matrix, while V_noises represents the strength of the noises learned by $\mathbf{W}_V$.



(a) $\alpha = 0.001$      (b) $\alpha = 0.01$      (c) $\alpha = 0.1$      (d) $\alpha = 0.2$

Figure 5: The heatmap of test loss on synthetic data across various *SNR*, N and label-flipping probability $\alpha$.

directed towards the signals. In contrast, during the noise learning phase in harmful overfitting, attention increasingly concentrates on the noises rather than the signals. Eventually, attention primarily focuses on the noises, causing the model to learn irrelevant information. We also observe that as the label-flipping noise $\alpha$ increases, a larger portion of the attention mechanism is directed towards the noises, leading the model to memorize more irrelevant information.

Figure 4 (a) and (b) demonstrate that the update of $W_V$ matrix can be characterized by three distinct phases. According to Assumption (7), the $W_V$ matrix starts with relatively small values due to random initialization. As training progresses, the model prefers to learn the signals rather than memorize the noises in benign overfitting, which is referred to as the signal learning phase. After a certain period, $W_V$ stops learning noises and V_noises converges to a constant, while $W_V$ continues to learn signals. In contrast, as training progresses, $W_V$ prefers to learn noises in harmful overfitting and $W_V$ completely memorizes noises ultimately. Furthermore, we observe that the $W_V$ matrix memorize more noises as the labeled noise $\alpha$ increases when benign overfitting occurs.

We further conduct experiments on two types of overfitting test errors as shown in Figure 1, providing empirical verification for our theoretical results in Theorem 1 and Theorem 2. When benign overfitting occurs, the initialization stage is brief, and the test loss remains at a significantly high value. The model gradually learns the signals, with the test loss decreasing rapidly until it reaches the $\epsilon$ level. During the convergence phase, the test loss stabilizes at the $\epsilon$ level. In contrast, when harmful overfitting occurs, the model prefers to learn noises, with the test loss increasing rapidly and eventually diverging.

## 5.2 TRANSITION BETWEEN BENIGN OVERFITTING AND HARMFUL OVERFITTING

As illustrated in Figure 5, there is a clear distinction between benign overfitting and harmful overfitting under varying labeled noise $\alpha$ and SNR. The test loss shows a decreasing trend as both N and SNR increase. To further explore this transition, we apply additional processing based on Figure 5. Figure 6 shows that the boundary does not undergo any significant spatial deformation as $\alpha$

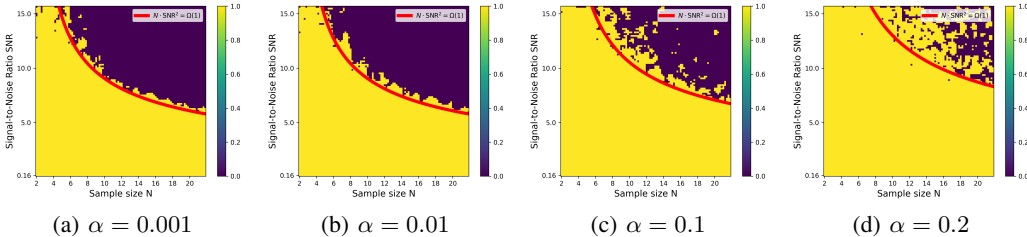

(a) $\alpha = 0.001$      (b) $\alpha = 0.01$      (c) $\alpha = 0.1$      (d) $\alpha = 0.2$

Figure 6: Under varying labeled noise $\alpha$, benign overfitting is depicted in yellow, while harmful overfitting is shown in purple. The transition between two types of overfitting is illustrated by a red curve.

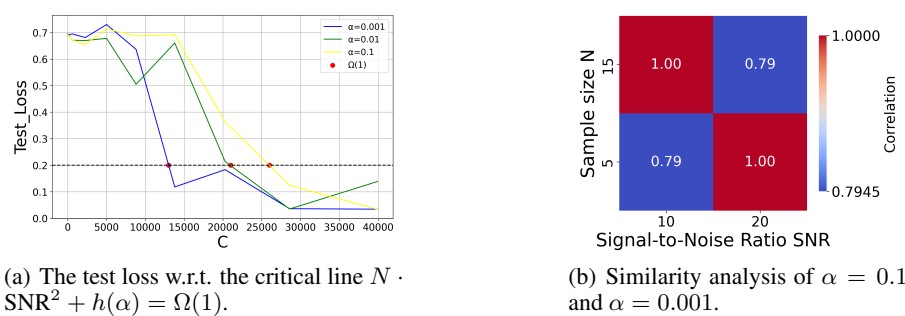

(a) The test loss w.r.t. the critical line $N \cdot \mathrm{SNR}^2 + h(\alpha) = \Omega(1)$.

(b) Similarity analysis of $\alpha = 0.1$ and $\alpha = 0.001$.

Figure 7: (a) shows the variation of $\Omega(1) - h(\alpha)$ with $\alpha$, while (b) shows the similarity between the two conditions $\alpha = 0.1$ and $\alpha = 0.001$, with higher scores indicating higher similarity.

increases. Instead, it simply shifts spatially. This observation aligns with our theory, which indicates that the transition between benign and harmful overfitting is primarily governed by SNR and $N$, while $\alpha$ only influences the translation $h(\alpha)$.

### 5.2.1 THE IMPACT OF CRITICAL LINE

The boundary $N \cdot \mathrm{SNR}^2 + h(\alpha) = \Omega(1)$ represents the minimum condition under which benign overfitting occurs. As illustrated in Figure 7 (a), we show how $N \cdot \mathrm{SNR}^2$ varies with changes in $\alpha$. The figure clearly demonstrates that the likelihood of convergence toward $\Omega(1)$ shifts with changes in $\alpha$. Specifically, as $\alpha$ increases, the term $\Omega(1) - h(\alpha)$ rises, indicating a corresponding decrease in $h(\alpha)$. Furthermore, Figure 7 (b) reveals that the shape of the curve remains consistent, suggesting that $\alpha$ affects only the spatial displacement $h(\alpha)$ of the curve, without altering its overall form. The curves for $\alpha = 0.1$ and $\alpha = 0.001$ demonstrate a striking similarity, which verifies our theory: varying $\alpha$ does not impact the distribution of the data; instead, it only influences the spatial offset $h(\alpha)$ of the boundary line.

## 6 CONCLUSION AND FUTURE WORK

This paper studies the training dynamics, convergence, and generalization of a two-layer transformer with labeled noise. Firstly, we present generalization error bounds for both benign and harmful overfitting under varying signal-to-noise ratios (SNR). Secondly, we categorize the training dynamics into three stages and provide corresponding stage-wise error bounds. One limitation of our study is that the transformer model we analyze consists of only two layers. The more complex softmax and multi-layer attention mechanisms in deeper transformers create significant challenges in separating signal from noise, complicating the analysis of their training dynamics and generalization. An important direction for future work is to extend our analysis to deeper architectures.

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
