# OpenReview forum: "Understanding Generalization in Transformers: Error Bounds and Training Dynamics Under Benign and Harmful Overfitting"
_ICLR.cc/2026/Conference — Submitted to ICLR 2026_

### Official Review · Reviewer_3Ed7 · 2025-10-16

**Soundness:** 1
**Presentation:** 2
**Contribution:** 1
**Rating:** 0
**Confidence:** 3

**Summary:**

Benign and harmful overfitting have been extensively studied in the past few years in many settings and models. More recently, there has been interest in analyzing benign overfitting in simple transformers. This work aims to extend the previous works on benign/harmful overfitting in transformers by providing an analysis for a more realistic setting. They also provide experiments to support their theory.

**Strengths:**

The paper considers an interesting problem, and focuses on a setting that is more realistic and complex than prior works. Specifically, they consider label noise, both key and query matrices in the softmax, all matrices (including the value matrix) are trainable, there is self-attention, they consider many steps of gradient descent, and consider both benign and harmful overfitting.

**Weaknesses:**

My main concern is that, as far as I understand, the proofs are incorrect.

First, the analysis relies on the equations from Appendix E, which are taken from Jiang et al.  (2024). However, the setting of Jiang et al. is different from the one in the current paper. A main difference is that Jiang et al. did not have label noise. Also, Jiang et al. considered noise tokens that are orthogonal to the signal tokens, and Jiang et al. did not consider a model of the form $f_{+1}-f_{-1}$ as in the current work. Hence, it is not clear that the equations from Jiang et al.’s analysis also hold here. The current paper uses these equations without providing a proof or an explanation. The authors do not analyze the dynamics of GD directly; they merely rely on the equations from Jiang et al., which might not be relevant here.

Moreover, there are claims that are informal or unclear. For example:
- It is not clear what the $\approx$ in the last displayed equation on page 19 means.
- The analysis of stage one is essentially an analysis of the random initialization, but why does it hold for every $t \leq T_1$ for the appropriate $T_1$?
- On page 21, it is claimed that the inner product between $\xi$ and $Wv$ has a normal distribution, but $W$ depends on $\xi$, so it’s not necessarily the case. Also, when analyzing the expectation of $g(\xi)$ (see Eq. 29), the authors seem to bound the expectation of the absolute value of $g(\xi)$ (otherwise, according to their aforementioned claim about the distribution of g(\xi) the expectation should be zero).

Hence, I recommend “strong reject”.
If I misunderstood something, I will be happy to reconsider my evaluation.

**Questions:**

See the “weaknesses” paragraph.

---

### Official Review · Reviewer_ADv3 · 2025-11-01

**Soundness:** 4
**Presentation:** 4
**Contribution:** 4
**Rating:** 10
**Confidence:** 4

**Summary:**

This is a theoretical work on (small) transformers and benign/harmful overfitting.
The authors analyze the generalization error and find the behavior to be grouped into three stages of training.
They also back up their theory with experiments that cohere with the theoretical prediction.
The theory is for data that are length-2 sequences where the tokens are drawn from a gaussian mixture model with orthogonal centroids. The first token is the signal and the second token is purely noise.

**Strengths:**

For benign overfitting, they have a fine-grained analyze of the train/test error versus gradient descent iteration number.
The data generation process allows label noise: alpha controls the error flipping probability.

It is interesting that they distinguish an early signal learning phase, and a late convergence phase. The early signal learning phase is presumbly where most test performance improvement is made. But the late-stage analysis also reveals that well-tuned initialization can help make further improvements in late training (i.e., when training loss is driven to zero).

Under a different setting of SNR and the noise flipping constant, the authors show harmful overfitting. It's nice to have both sides of the overfitting.

The experimental results are well-constructed and in my opinion faithfully follow the theory (Figure 6 in particular).

**Weaknesses:**

I think this is a really solid theory paper, so i don't have any major concerns. The author already discuss many future directions that one might call "weakness". But I think the paper is valuable contribution to research as is, i.e., it highlights how attention score shifts depending on SNR. One thing I think the author may want to add is remark concerning weakening the orthogonality condition.

**Questions:**

How important is the orthogonality condition?

How important the assumption that the sequences are length 2?

From my understanding, benign overfitting is not as relevant in the practice of language model training (unlike say learning to classify imagenet), since one doesn't take as many passes over the training data. can the authors comment about this?

---

### Official Review · Reviewer_VQ35 · 2025-11-02

**Soundness:** 1
**Presentation:** 1
**Contribution:** 2
**Rating:** 2
**Confidence:** 2

**Summary:**

The paper presents a theoretical study of a two-layer nonlinear transformer under label-flip noise, provides stage-wise error bounds for two regimes (benign vs harmful overfitting), and supports claims with synthetic experiments that visualize the three training phases.

**Strengths:**

The paper addresses an important problem: specifically, attempting to better understand the generalization property of transformers under label noise, with respect to both benign and harmful overfitting.

**Weaknesses:**

**W1.** In the introduction, the authors claim to have performed extensive experiments to validate their theoretical findings. However, in practice, they only conduct experiments on simulated data without employing any real-world benchmarks that deal with varying labeled noise (there are many available, for instance, on CIFAR). Furthermore, the training iterations (max 500) and sample sizes (max 20) are both very small, underlining the toy empirical setup.

**W2.** The paper can be improved in terms of clarity and presentation. Please see the following examples:

- Currently, captions of figures (Fig. 1-4) only loosely describe what they represent in short on-liner captions without providing more nuanced takeaways.

- Contribution 1, i.e., just "considering a nonlinear transformer with softmax activation function" cannot be considered as a contribution. It should be better reframed.

- There is no reference to any proof for Theorems 1 and 2 in the Appendix. ICLR allows for the Appendix to be placed after the references.

- The authors claim that focusing on benign overfitting in transformers is more challenging than in neural networks, but transformers are neural networks, and there is no explanation for this claim.

**W3.** The theoretical upper bounds for Phases 1 and 2 (Figure 2) appear to be quite inaccurate as an estimate compared to the empirical loss.

**W4.** The number of references appears quite limited for a main conference paper addressing a popular problem, such as generalization under label noise, raising doubts regarding the coverage of the related literature.

**Questions:**

Please see Limitations.

---

### Official Review · Reviewer_NCWG · 2025-11-03

**Soundness:** 2
**Presentation:** 2
**Contribution:** 2
**Rating:** 2
**Confidence:** 1

**Summary:**

The paper analyzes a two‑layer transformer with softmax self‑attention trained by gradient descent on logistic loss and evaluated with 0–1 test error under label‑flipping noise (flip rate α).  The authors give stage‑wise (Phase 1/2/3) error bounds for benign and harmful overfitting and propose a critical condition.

**Strengths:**

Tackles nonlinear self‑attention with label noise and analyzes both benign and harmful overfitting in one framework; the comparison table highlights coverage vs. prior work (Table 1, p.2).
Provides stage‑wise, time‑dependent bounds; in benign Phase 2/3 the exponents scale, and Phase 3 explicitly depends on σ_V (Theorem 1, p.6).
Clear linkage between dynamics and generalization: attention ratios and value components evolve as predicted (Figs. 3–4, pp.7–8).

**Weaknesses:**

The proof seems to require a lot of revision for soundness and clarity (See Questions).

**Questions:**

Q.1. Theorem 14 yields only a constant‑level bound during initialization. Given your own derivation on pp.19–20 (uniform attention at init; f(X,theta(0)) is centered Gaussian), one can already conclude L_D^{0-1}(theta(0)) ≤ alpha + 1/2 from symmetry, without an unspecified O(1). As a result the theorem, in its current form, appears nearly vacuous for 0–1 loss. Please either (i) sharpen it to an explicit non‑trivial bound, or (ii) demote it to a lemma/remark and reserve theorem status for the stage‑wise bounds that contain informative time/SNR/sigma_V dependencies.

Q.2. Eq. (27) proves P{ y_b f ≥ log(1+exp(−1/2)) } ≥ 1/2, but the proof of Theorem 20 uses the larger threshold log((1+2 exp(1/2))/(1+exp(1/2))). However, log(1+exp(−1/2)) M< log((1+2 exp(1/2))/(1+exp(1/2))). so the latter is strictly larger. Since probabilities are monotone in the threshold, Eq. (27) does not imply the ≥1/2 statement at the bigger threshold; hence the proof is incomplete. Please either align Theorem 20’s threshold with Eq. (27), or introduce a new lemma showing that at least half of the mass lies above the larger threshold (e.g., via a median or tail bound argument consistent with your stage‑wise analysis). See p.19 (Eq. 27) vs. p.29 (Theorem 20).

Q.3. Theorem 18 adds the approximate argument \approx too easily. A more rigorous discussion is needed. Also, A, B, C, D, E, and F are not independent because they contain multiple common variables. They cannot be analyzed independently and simply added together. The correlation between each must also be considered.

Q.4. Equations (5)–(10) and (18)–(23) are imported "According to Jiang et al. (2024)," but Jiang et al. analyze the clean‑label case; in your noisy‑label setting (alpha in [0, 1/2)), shouldn’t these dynamics be re‑derived to show explicit alpha‑dependence, or otherwise justified as still valid upper/lower bounds without change (E.1/E.2; Def. 1 on p.4)?

The paper’s proofs hinge on E.1 (5–10: growth of Lambda and softmax ratios in Phase 2) and E.2 (18–23: V‑updates and attention behavior on the harmful side), yet these inequalities contain no alpha terms even though label flipping is introduced (Def. 1). As written, they read as a direct reuse of clean‑label results. Please either add a "gradient alignment under noise" lemma that propagates alpha into the growth factors (e.g., constants scaled by \approx 1−2*alpha), or explain why the clean‑label dynamics remain valid as bounds under your assumptions.

 Under the stated training (logistic loss; p.5), the conditional expected gradient satisfies E_y[nabla_theta ell(y f)] = (alpha − sigma(− y_hat*f)) * y_hat * nabla_theta f, so it becomes anti‑aligned whenever sigma(− y_hatf) ≤ alpha, i.e., y_hat*f ≥ M_alpha with M_alpha := log((1−alpha)/alpha). In this regime the signal–noise logit gap may stop increasing, attention can shift to noise, and V_xi terms grow—matching your harmful Phase‑2/3; please indicate how this threshold is reflected in (5)–(10) and (18)–(23), or revise those steps accordingly.

Q.5. In the initialization phase you treat the model score f(theta(t), X) as (approximately) Gaussian to prove Lemmas 14 and 17. Does this assumption remain valid beyond t=0, i.e., throughout Phase 1 after several gradient steps? After the first updates, f(theta(t), X) is a nonlinear functional of Gaussian noise through softmax attention and time‑varying weights, so its distribution need not stay Normal.

Q.6. Jiang et al. (2024) also analyze training in three stages. In what sense is your Stage‑1/2/3 decomposition new?
 Please clarify whether the novelty lies in. If the stage concept itself is not new, please reframe the contribution claims (abstract/introduction and comparison table) to emphasize the quantitative, noisy‑label advances over Jiang et al.

---

### Meta-Review · Area_Chair_6GSZ · 2026-01-02

**Summary:**

Multiple reviewers discovered errors in the proof which seem insurmountable - the authors used technical lemmas about the dynamics of G.D.-trained transformers from Jiang et al., but Jiang et al.'s distributional setting did not include label-flipping noise.  The introduction of label-flipping noise is essential to the investigation of benign overfitting, and in general will result in very different learning dynamics than without, and is indeed the central difficulty with investigating the training dynamics of overfitting.  There were also a number of places where informal $\approx$ symbols were used, when a formal, complete proof was needed.

**Reviewer Concerns:**

I concur with the multiple reviewers' assessment of the inaccuracy of the results.

**Reviewer Scores:**

I think it is likely that Reviewer ADv3 would have significantly reduced their score if they had participated in the discussion as they did not catch the error.  The other reviewers would have remained the same (with scores <= 2).

---

### Decision · Program_Chairs · 2026-01-26

Reject